# Structural basis for IL-33 recognition and its antagonism by the helminth effector protein HpARI2

Abhishek Jamwal [1,2], Florent Colomb[3], Henry J. McSorley [3] & Matthew K. Higgins [1,2]

IL-33 plays a significant role in inflammation, allergy, and host defence against parasitic helminths. The model gastrointestinal nematode *Heligmosomoides polygyrus bakeri* secretes the Alarmin Release Inhibitor HpARI2, an effector protein that suppresses protective immune responses and asthma in its host by inhibiting IL-33 signalling. Here we reveal the structure of HpARI2 bound to mouse IL-33. HpARI2 contains three CCP-like domains, and we show that it contacts IL-33 primarily through the second and third of these. A large loop which emerges from CCP3 directly contacts IL-33 and structural comparison shows that this overlaps with the binding site on IL-33 for its receptor, ST2, preventing formation of a signalling complex. Truncations of HpARI2 which lack the large loop from CCP3 are not able to block IL-33-mediated signalling in a cell-based assay and in an in vivo female mouse model of asthma. This shows that direct competition between HpARI2 and ST2 is responsible for suppression of IL-33-dependent responses.

Interleukin-33 (IL-33) belongs to the IL-1 cytokine family and is an inducer of host allergic and inflammatory responses[1]. Full-length IL-33 is stored in the nucleus of a range of cells, bound to heterochromatin[2,3]. One of the main signals for IL-33 release into the extracellular environment is cellular damage or necrosis caused by factors including allergens, smoking or tissue injury. Once released from the damaged cell, IL-33 functions as an endogenous danger signal or alarmin, triggering an immune response[2,4]. As IL-33 is upregulated in human asthma and allergy, as well as in related animal models, it is a promising target for therapeutics against allergic inflammatory diseases[5].

The molecular mechanism by which IL-33 mediates signalling is well characterised[6]. Extracellular IL-33 binds to its cognate receptor, the membrane-bound form of ST2[4,7,8]. The IL-33-ST2 complex then recruits the co-receptor, IL-1 Receptor Accessory Protein (ILRAcP), forming an active signalling complex, which has been structurally characterised[9]. Immune cells, such as eosinophils, basophils, mast cells, macrophages, Th2 cells and group 2 innate lymphoid cells (ILC2s), respond to IL-33-mediated signalling, leading to production of pro-inflammatory cytokines[5]. IL-33 is released in a reduced active form which is rapidly oxidised into an inactive form, in a process thought to limit signalling to occur only close to the site of release[10].

IL-33 is also central to the immune response in helminth infections[1]. Mice deficient in IL-33 or ST2 are more susceptible to a range of helminths but are protected from allergic pathology in models of asthma[1,2]. Therefore, suppression of the IL-33 pathway is an effective mechanism by which helminths could avoid ejection by the immune system and would have the side-effect of suppressing allergic responses. Indeed, helminth infections have been linked with immunosuppression in human populations and have been proposed to reduce the prevalence of allergic diseases such as asthma[11]. In animal models, infection with these parasites reduces pathology related to allergy[12].

A direct link between parasite suppression of IL-33 responses and reduced allergic reactions has been identified for the mouse-infective helminth *Heligmosomoides polygyrus bakeri*[13]. *H. polygyrus bakeri* secretes a cocktail of molecules, known as the *H. polygyrus bakeri*

[1]Department of Biochemistry, University of Oxford, South Parks Road, Oxford OX1 3QU, UK. [2]Kavli Institute for Nanoscience Discovery, Dorothy Crowfoot Hodgkin Building, University of Oxford, South Parks Rd, Oxford OX1 3QU, UK. [3]Division of Cell Signalling and Immunology, School of Life Sciences, University of Dundee, Dow Street, Dundee DD1 5EH, UK. ✉e-mail: HMcsorley001@dundee.ac.uk; matthew.higgins@bioch.ox.ac.uk

excretory/secretory products (HES), which have a range of immuno-modulatory activities[14–16]. HpARI and HpBARI are secreted families of proteins which bind to IL-33 and ST2 respectively, modulating IL-33-mediated signalling and downstream immune cell activation[15,16]. The HpARI family consists of HpARI1, HpARI2 and HpARI3, of which HpARI2 is the best characterised[15,17,18]. HpARI2 binds to both IL-33 and genomic DNA in necrotic cells, allowing it to tether IL-33 within dead cells. In doing so, HpARI2 effectively blocks IL-33 responses in acute and chronic models of asthma and when administered in models of other nematode infections[15,19–21]. HpARI2 is composed of three CCP domains, with CCP1 implicated in DNA binding, while CCP2 and CCP3 are implicated in IL-33 binding[15,18]. Intriguingly, while full-length HpARI2 blocks ST2-mediated signalling, a form lacking CCP3 stabilises IL-33 and potentiates its activity[18].

The central role of IL-33-mediated signalling in allergic inflammatory responses has led to substantial interest in developing modulators of IL-33 activity for therapeutic use[22]. Indeed, monoclonal antibodies which block IL-33 signalling have been shown in animal models and clinical trials to reduce allergic inflammation and asthmatic pathology[23–26]. With IL-33 binders able to both potentiate and suppress IL-33-mediated activity, this brings risks, as potentiation of IL-33 signalling may lead to increased allergic reaction[18]. It is therefore important to understand the molecular mechanisms through which both potentiation and suppression operate. However, no structural studies exist which reveal how IL-33 binders can modulate its activity. To understand how HpARI2 alters IL-33 activity, and how HpARI2 truncations can either activate or suppress IL-33 function, we therefore aimed to determine the molecular mechanism of IL-33 modulation by HpARI2. Here, we show, through structural and functional studies, that CCP domains 2 and 3 of HpARI interact with IL-33. We find that a long loop emerging from CCP3 sterically prevents IL-33 from binding to its receptor, ST2, directly inhibiting IL-33-mediated signalling.

## Results

### The CCP1 domain of HpARI2 is not required for inhibition of IL-33-mediated signalling

Previous studies revealed that HpARI2 is a modular protein made of three CCP-like domains (CCP1-3) (Fig. 1a), which interacts with mouse (m)IL-33, both in its full-length form and after truncation of CCP1 or CCP3 (HpARI2_CCP1/2 and HpARI2_CCP2/3)[15,18]. To determine the structure of the active region of HpARI bound to IL-33, we first confirmed which regions of HpARI are required for IL-33 binding and inhibition, using quantitative surface plasmon resonance (SPR) and cell-based inhibition assays. We first used size exclusion chromatography (SEC) to show that the truncated HpARI2s, like their precursor protein, form stable binary complexes with mIL-33 (Supplementary Fig. 1). We then proceeded to measure binding affinities and kinetics for these interactions using SPR. Full-length HpARI2 bound to mIL-33 with high affinity ($K_D \sim 48$ pM) (Fig. 1b, top panel and Supplementary Table 1). However, truncation of CCP3 reduced the affinity by a thousand-fold ($K_D \sim 49$ nM), with the complex dissociating more rapidly with a $k_{off}$ at least two orders of magnitude higher than that of full-length HpARI2 (Fig. 1b, middle panel and Supplementary Table 1). Truncation of CCP1 had a smaller effect on the affinity, with a $K_D$ of ~4 nM (Fig. 1b lower panel and Supplementary Table 1). Therefore, all three CCP domains contribute to mIL-33 binding, with CCP1 making the least contribution.

We next assessed the impact of HpARI2 and its truncated derivatives on IL-33-mediated activation of ST2-positive mouse type 2 innate lymphoid cells (ILC2), using secretion of IL-5 as a readout of ILC2 activation (Fig. 1c). We found that both full-length HpARI2 and HpARI2_CCP2/3 fully inhibited IL-33-mediated IL-5 production in a dose-dependent manner, whereas ILC2 activation by mIL-33 was unaltered by HpARI2_CCP1/2 in this assay (Fig. 1d). We observed a close match between the concentrations required for inhibition of IL-33

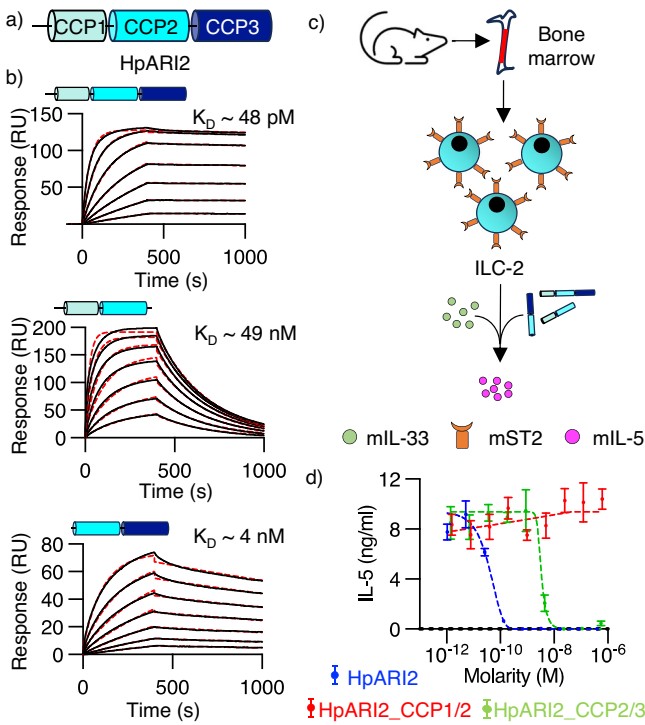

**Fig. 1 | The CCP1 domain does not contribute to the effector function of HpARI2. a** Schematic representation of the three CCP-like domains of HpARI2 (CCP1-3). **b** Surface plasmin resonance analysis of the binding of full-length HpARI2 (upper panel), HpARI2_CCP1/2 (middle panel) and HpARI_CCP2/3 (lower panel) (red dashed lines) to mIL-33 were generated by flowing 2-fold serial dilutions of mIL-33 starting from 40 nM (over HpARI2), 200 nM (over HpARI2_CCP2/3), and 1000 nM (over HpARI2_CCP1/2) over surfaces immobilised with HpARI2 truncations. Indicated $K_D$ values were deduced from Langmuir fit (red dashed lines) and the experiments were conducted twice ($n = 2$). **c** Overview of the cellular assay for assessing effector function of HpARI2s. Mouse bone marrow cells were isolated and cultured with IL-2, IL-7, and IL-33 in the presence of HpARI2 truncations at various concentrations, followed by measurement of mIL-5 levels in cultured supernatants. **d** Concentration-response curves for the inhibition of signalling by various HpARI2 forms as determined by IL-5 production. Error bars indicate mean +/- standard error of the mean of 4 technical replicates. Data are representative of 3 repeat experiments ($n = 3$). Source data are provided as a Source Data file.

function by HpARI2 ($EC_{50} = 40$ pM) and HpARI2_CCP2/3 ($EC_{50} = 3$ nM) and their affinities for IL-33, suggesting that HpARI inhibits through a direct mechanism. From this data we conclude that the last two CCP modules of HpARI2 are sufficient to inhibit IL-33-function, guiding our structural studies.

### Structural characterisation of the HpARI2_CCP2/3:mIL-33 complex

We next attempted to obtain crystals of mIL-33 bound to our different HpARI2 constructs. While crystals containing full-length HpARI2 did not form, crystals were obtained containing the HpARI2_CCP2/3:IL-33 complex and we determined the structure to 2.1 Å. (Fig. 2, Supplementary Table 2, Supplementary Fig. 2). The asymmetric unit of the crystal contained one copy of each of HpARI2_CCP2/3 and mIL-33. Each HpARI2_CCP2/3 molecule contacted two IL-33 molecules in the crystal (Supplementary Fig. 3a). To determine which of these represents the physiological complex, we identified HpARI2 residues within each interface and designed mutations to insert N-linked glycans at these sites. A glycan inserted in interface A (residue 152) did not affect binding of HpARI2 to IL-33 (Supplementary Fig. 3). In contrast, glycans inserted into interface B (residues 69 and 70) reduced IL-33 binding, as measured by SPR, despite demonstration of correct folding and glycan

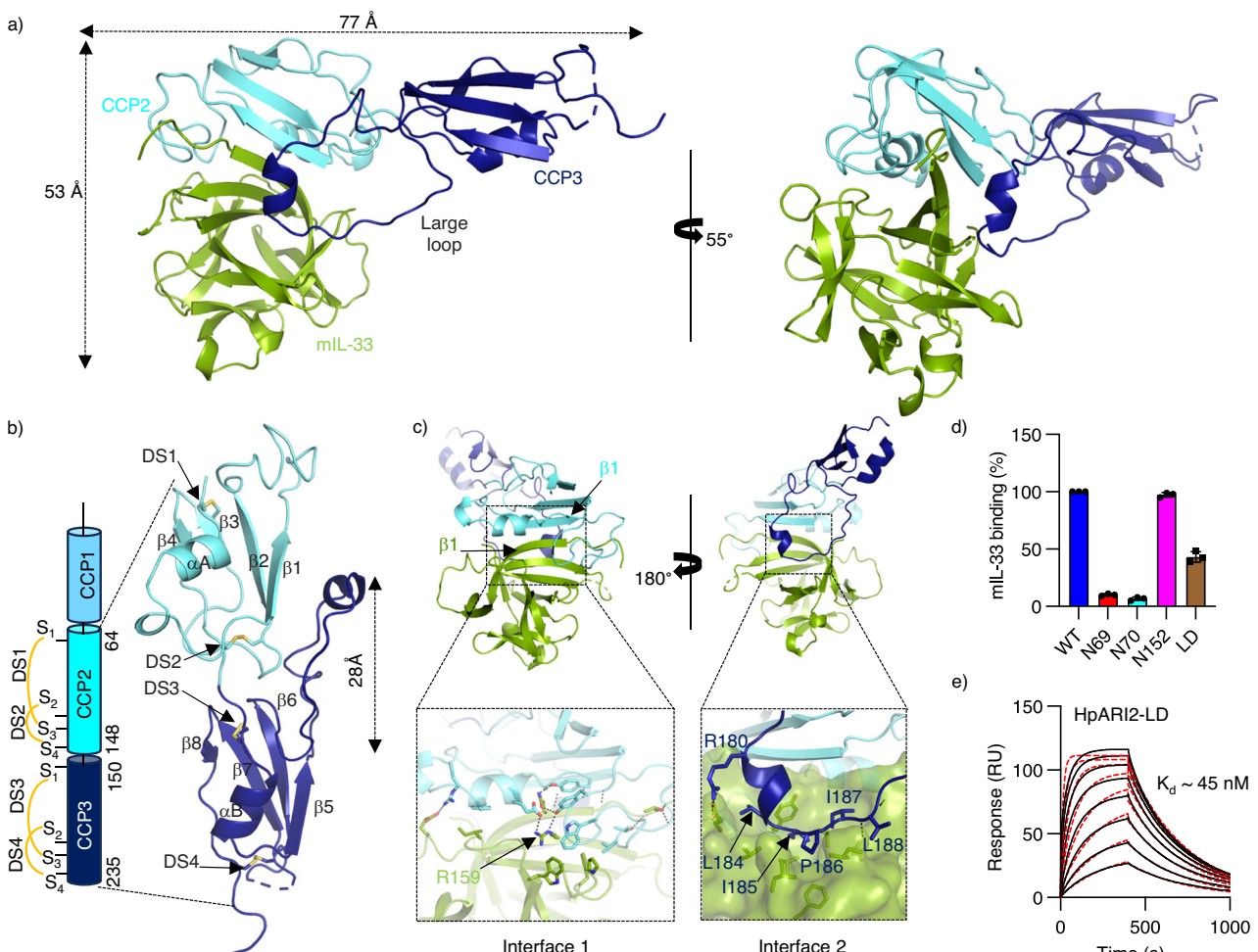

**Fig. 2 | The structure of the HpARI2_CCP2/3:IL-33 complex. a** Two views of HpARI2_CCP2/3 (CCP2 in cyan and CCP3 in blue) bound to mIL-33 (green). **b** The structure of HpARI2_CCP2/3 (CCP2 in cyan and CCP3 in blue) with disulphide bonds (DS1-4) shown as yellow sticks. The domain schematic on the left gives an overview of the structure with precise domain boundaries and patterns of disulphide bonds within each domain derived from the crystal structure. **c** Two views of the HpAR-I2_CCP2/3:mIL-33 complex with boxes below showing zoomed views of these interfaces. Interfacial residues are shown as sticks and hydrogen or polar interactions are indicated with a dashed line. **d** Binding of mIL-33 to HpARI2 variants determined by surface plasmon resonance analysis at a HpARI2 concentration of 100 nM. Bars represent the average and standard deviation for three independent measurements ($n = 3$). **e** Surface plasmon resonance analysis of HpARI2-LD binding to mIL-33. Multicycle response curves for two-fold mIL-33 dilutions starting from 2 μM ($n = 2$). Source data are provided as a Source Data file.

modification (Supplementary Fig. 3 and Supplementary Table 1). Therefore, HpARI2_CCP2/3 is an elongated molecule which uses both CCP domains to interact with IL-33 (Fig. 2a, Supplementary Fig. 2).

The structure of HpARI2_CCP2/3 revealed that the two CCP domains, which are joined by a 3 residue-long linker, are not flexibly associated but instead interact with a buried surface area of 522 Å² (Fig. 2b). Both domains adopt a CCP domain architecture, being formed from a 3 or 4-stranded β-sheet packed against a short helix and with two disulphide bonds (Fig. 2b). The domains differ in the length and organisation of their loops. In particular, the 28 residue-long loop which connects strands β5-β6 of CCP3 extends ~28 Å away from CCP3 towards the base of CCP2 (Fig. 2b). Structural similarity analysis using DALI revealed that this loop is absent among all other CCP domains of known structure[27]. SAXS data for both HpARI2_CCP2/3 and full-length HpARI2, alone and in complex with IL-33, shows that these molecules adopt a similar conformation in solution as that observed when IL33 bound, or when in the crystal structure (Supplementary Fig. 4). Therefore, unlike ST2, which folds around IL-33, HpARI appears to adopt the correct conformation for binding in the absence of ligand. The SAXS data, together with AlphaFold2 prediction, also suggest that CCP1 also forms a substantial interaction interface with CCP2, forming an elongated arrangement (Supplementary Fig. 4).

We next analysed how HpARI2_CCP2/3 recognises mIL-33. In the complex, mIL-33 adopts a conformation similar to that previously observed for unbound human IL-33 (root mean square deviation of 0.904 Å for 102 residues) and of the mouse IL-33 structure bound to ST2 (root mean square deviation of 0.64 Å of 104 residues) (Supplementary Fig. 4e). IL-33 interacts with both CCP domains of HpAR-I2_CCP2/3 with a total buried surface area of 1548 Å². Most of the interaction occurs between the CCP2 domain and the rim of the mIL-33 β-barrel, with a total buried surface area of 1040 Å² and with contacts involving the edges of β-sheets and interacting side chains (Fig. 2c, left panel). The β-sheet contacts are mediated by main chain hydrogen bonds between CCP2-β1 and β1 of IL-33, which extends the CCP2 domain β-sheet (Fig. 2c, left panel). Residues from the outer β-barrel region of mIL-33 and from the α-helix and the loop connecting β1 and β2 strands of the CCP2 module stabilise the interface via both polar and non-polar interactions (Fig. 2c, left panel). Residue R159 of IL-33 lies at the centre of this interface and the R159A mutation causes a ~1000-fold reduction in binding affinity for HpARI2_CCP2/3 (Supplementary Fig. 5 and Supplementary Table 1). In contrast, mutating HpARI2 residues E70, Y112 and W107, which are also at the interface, had little effect, consistent with the large interface area (Supplementary Fig. 5 and Supplementary Table 1).

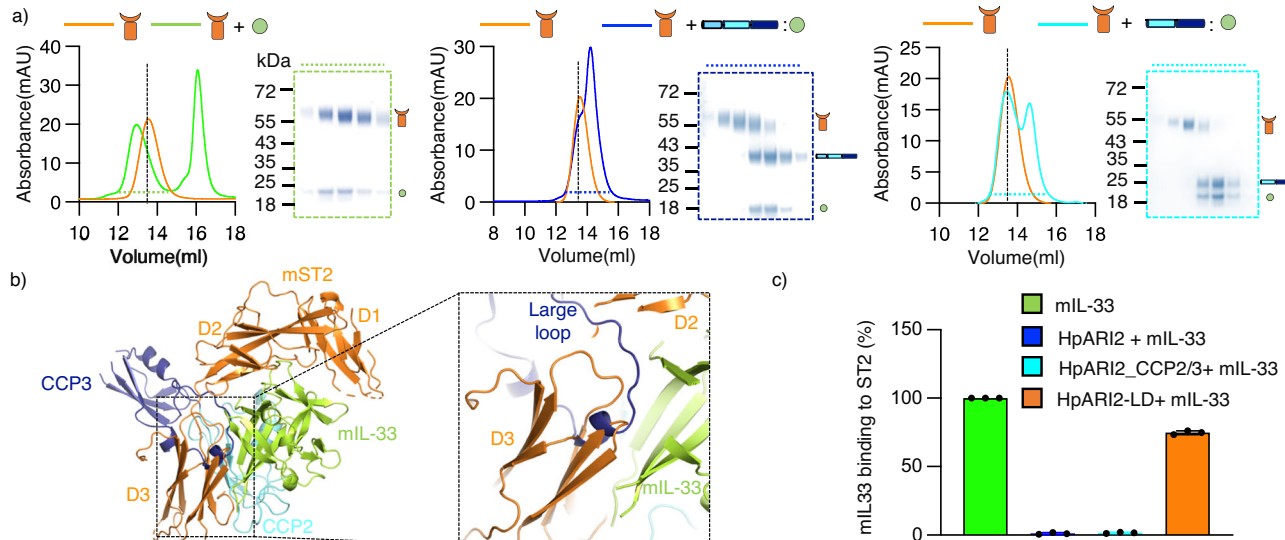

**Fig. 3 | HpARI2 competes for binding to IL-33 with the receptor ST2.**
**a** Assessment of the binding of the mST2 ectodomain (orange curve) to mIL-33 (green), HpARI2:mIL-33 (blue) and HpARI2_CCP2/3:mIL-33 (cyan) in a size exclusion column. The SEC curves are representative of two independent experiments (*n* = 2). Coomassie-strained gels show fractions highlighted on the size exclusion traces by dotted blue lines. **b** Overlay of the HpARI2_CCP2/3:mL33 crystal structure on that of the mIL-33:mST2 complex (5VI4 (IL33:ST2:IL-1RAcP)), revealing a steric clash

between CCP3 and ST2. ST2 is orange, mIL-33 is green and the two CCP domains of HpARI2 are cyan (CCP2) and blue (CCP3). The dotted box to the right shows a close view of the complex. **c** Measurement by surface plasmon resonance of the binding of mIL-33 to immobilised ST2-Fc in the presence of HpARI2 and variants. Bars show the mean from three independent measurements (*n* = 3) with error bars showing standard deviation. Source data are provided as a Source Data file.

The CCP2 domain is also present in a truncated version of HpARI2 which stabilises IL-33 thereby potentiating its systemic effects[18]. The close match and extensive interface formed between HpARI2 and IL-33 may allow HpARI2 to limit the conformational change which occurs on IL-33 oxidation, allowing it to stabilise IL-33 in its active conformation (Supplementary Fig. 4e). We also collected SAXS data for HpAR-I2_CCP2/3 and full-length HpARI2 bound to IL-33. These are both consistent with the crystal structure. While this analysis places CCP1 close to IL-33, CCP1 does not make unambiguous direct contacts with IL-33 (Supplementary Fig. 4). The increase in affinity for IL-33 which occurs due to the presence of CCP1 is therefore more likely to be an indirect effect in which CCP1 stabilises the conformation of CCP2 in its binding to IL-33.

While humans are not a natural host for *H. polygyrus bakeri*, we also investigated the binding of HpARI2 to human IL-33 (Supplementary Fig. 6). This might be relevant if variants of HpARI2 are used to regulate IL-33 signalling in future therapeutics. We found that HpARI2 binds to human IL-33 with an affinity of 65 nM, which is more than a thousand-fold weaker than its binding to mouse IL-33 (Supplementary Fig. 6b). The alignment of mouse and human IL-33 molecules revealed a polymorphism at position 158, within the HpARI2 binding site, and a S158R mutant of human IL-33, which places the mouse specific residue in this location, increased HpARI2 binding by more than 100-fold (Supplementary Fig. 6b). Therefore, the same interface which is used by HpARI2 to bind mouse IL-33 can also bind to human IL-33 (Supplementary Fig. 6c).

All contacts between IL-33 and HpARI CCP3 are mediated by the long CCP3 loop. This adopts an extended conformation, with residues 180-184 forming a small helical turn at the tip (Fig. 2c, right panel). Interactions with IL-33 are mostly formed by non-polar residues from, and C-terminal to, a small helix which docks into a small concave, hydrophobic pocket on the top of the mIL-33 β-barrel (Fig. 2c, right panel). We next assessed the importance of this interaction by deleting residues 180-188 from full-length HpARI2 (HpARI2-LD), which will truncate the loop sufficiently to prevent it from reaching its binding site. This reduced mIL-33 binding by ~ 60%, compared with the more than 90% reduction observed for the glycosylation mutants at position

69 and 70 (Fig. 2d). Surface plasmon resonance measurements showed that HpARI2-LD bound to IL-33 with a $K_D$ of ~ 45 nM, which is ~1000-fold weaker than that of full-length HpARI2 (Fig. 2e and Supplementary Table 1). Indeed, the kinetic profile and $K_D$ for HpARI2-LD are similar to those of HpARI2_CCP1/2, supporting the hypothesis that this loop contains all of the binding determinants contributed by CCP3. Like-wise, when the glycosylation mutants at position 69 and 70, or HpARI2-LD were assessed for binding to human IL-33, they showed similar reduction or ablation of binding affinity for the cytokine, confirming that the same interface is used for binding of HpARI2 to mouse and human IL-33 (Supplementary Fig. 6c). Therefore, the structure reveals how domains CCP2 and 3 of HpARI2 each contribute to the interaction with IL-33. We next investigated how these interactions prevent IL-33-mediated signalling.

### HpARI2 blocks IL-33 from binding to its receptor ST2 through a steric mechanism involving the large loop of CCP3

We next investigated whether HpARI2 and its derivatives can prevent IL-33 from binding to its receptor, ST2. We started by demonstrating, using a SEC assay, that IL-33 and ST2 co-migrated as a complex (Fig. 3a). In contrast, preincubation of IL-33 with HpARI2 or HpAR-I2_CCP2/3 blocked the formation of the IL-33:ST2 complex, instead forming complexes between IL-33 and HpARI2 derivates which migrate separately to the ST2 ectodomain (Fig. 3a). To rationalise this observation, we compared our crystal structure of the HpARI2_CCP2/3:mIL-33 complex with a previously determined structure of IL-33 bound to ST2 (Fig. 3b). ST2 binds to two distinct sites on mIL-33, with domains D1 and D2 of ST2 binding one site and D3 binding the second site[28]. Aligning our crystal structure of HpARI2_CCP2/3:mIL-33 with that of the mIL-33:mST2 complex, based on the published IL-33:ST2:ILRAcP structure (PDB ID = 5VI4 (IL33:ST2:IL-1RAcP)) reveals a clash between the large loop of HpARI2 CCP3 and domain 3 (D3) of ST2 (Fig. 3b and Supplementary Fig. 7a). To validate this, we used an SPR assay to measure the binding of mIL-33 binding to Fc-tagged mST2. We then studied this interaction in the presence of full-length HpARI2, HpAR-I2_CCP2/3 and the loop deletion, HpARI2-LD. While full-length HpARI2 and HpARI2_CCP2/3 abolished the IL-33-ST2 interaction, the same

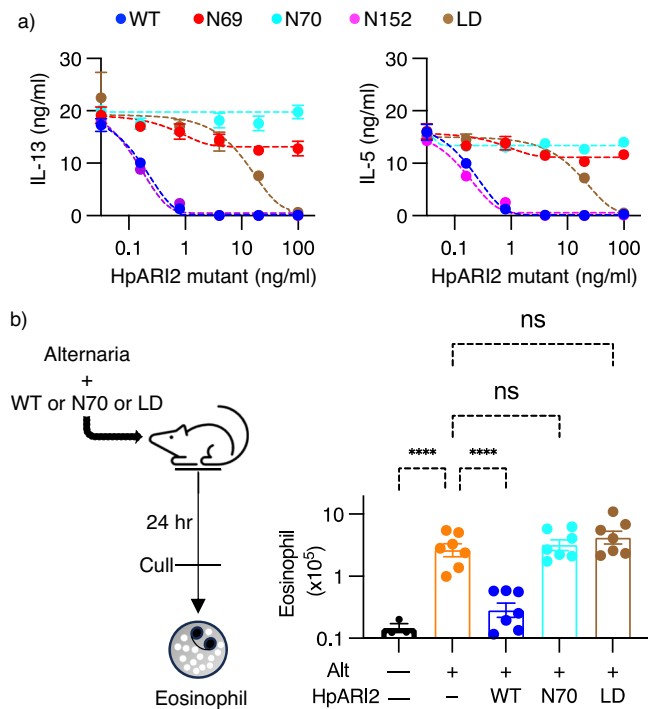

**Fig. 4 | Assessing the effect of HpARI2 mutants on IL-33 signalling and function. a** Mouse bone marrow cells were cultured for 5 days with IL-2 (10 ng/ml), IL-7 (10 ng/ml) and IL-33 (1 ng/ml) in the presence of a range of concentrations of HpARI2 or its N69, N70, N152 or LD mutants. Levels of IL-13 (left) and IL-5 (right) in culture supernatants were measured by ELISA. Error bars indicate mean +/- SEM of 6 technical replicates per condition. Data is representative of 3 repeat experiments. **b** The left panel is a schematic showing that Alternaria allergen (50 μg), together with HpARI2 WT, N70 or LD mutants (10 μg) were intranasally administered to BALB/c female mice. The right depicts calculated cell numbers of SiglecF$^{hi}$CD11c$^-$CD45$^+$ live cells in right lobes of lung. Data are pooled from 2 repeat experiments for $n = 7$, except for the PBS control which has $n = 3$. Error bars indicate mean +/- SEM. One-way ANOVA with Sidak's multiple comparisons test was used to compare each group to Alt control ****$p < 0.0001$, $ns = p > 0.05$. Source data are provided as a Source Data file.

concentration of HpARI2-LD only reduced ST2 binding by ~25 % (Fig. 3c and Supplementary Fig. 7b). Therefore, while the HpARI2 CCP1 domain is not required for inhibition of the IL-33-ST2 interaction the large loop emerging from CCP3 is important, as it directly competes with ST2. As HpARI prevents IL-33 from binding to ST2, it will prevent ST2 from adopting the correct conformation to bind to the ILRAcP co-receptor and thereby prevent IL-33-mediated signalling[9].

### HpARI2 inhibits IL-33-mediated signalling by directly blocking binding of ST2

We next assessed the effects of HpARI2 mutations in functional assays designed to measure IL-33-mediated signalling. In the first assay, we measured the production of cytokines IL-5 and IL-13 by bone marrow-derived ILC2s as an outcome of IL-33-mediated signalling. Addition of HpARI2 caused a complete dose-dependent inhibition of IL-5 and IL-13 release, demonstrating its inhibition of IL-33-mediated signalling (Fig. 4a). The HpARI2-N152 mutant, previously shown to not affect IL-33 binding, showed a similar dose response profile to full-length HpARI2 (Fig. 4a). In contrast, HpARI2-N69 and HpARI2-N70 did not reduce IL-5 and IL-13 secretion, consistent with a loss of IL-33 binding. HpARI2-LD required ~100-fold higher protein concentration than its wild-type precursor to achieve the same level of inhibition (Fig. 4a). Therefore, the effects of HpARI2 mutations on IL-33-mediated signalling mirrored their effects on IL-33 binding and blockage of the IL-33-ST2 interaction.

We next tested the mutants for their ability to suppress type 2 immune responses in an in vivo model, in which Alternaria allergen is intranasally administered to BALB/c mice in the presence of HpARI2 derivatives. The outcome was assessed by measuring eosinophil recruitment to the lungs 24 h after Alternaria administration (Fig. 4b). While administration of full-length HpARI2 reduced lung eosinophil cell numbers as previously shown, we observed no such suppression with either HpARI2-N70 glycan mutant or with HpARI2-LD (Fig. 4b and Supplementary Fig. 8). The equivalent effects of HpARI2 mutants in vitro with those observed in the mouse model show that the direct inhibition observed in the crystal structure is the mechanism underlying inhibition of IL-33-mediated signalling in vivo.

## Discussion

Our structure shows how HpARI2 interacts with IL-33, due to two distinct interfaces mediated by two different CCP domains. The first interface is mediated by the CCP2 domain of HpARI2, which provides more than two thirds of the bound surface area of the interaction. However, this interface does not overlap with the binding sites on IL-33 for its receptors, ST2 or ILRAcP[9]. As a HpARI2 truncation containing CCP2 but lacking CCP3 does not prevent IL-33 from binding to ST2, this interface is not sufficient to prevent signalling. The second HpARI2-IL-33 interface is entirely mediated by a long loop which emerges from domain CCP3. This is smaller in surface area but does contribute to binding affinity. It is also essential for the ability of HpARI2 to inhibit IL-33 from binding to ST2 due to a direct steric clash, as CCP3 and ST2 have overlapping binding sites on IL-33. Therefore, HpARI2 uses two different modules to bind to IL-33, one which provides much of the interaction affinity, but no inhibitory potential, and a smaller one which is required to block signalling.

What implications does this structural insight have for how we can regulate IL-33? HpARI2 blocks signalling by preventing ST2 binding[15] while deletion of the third CCP domain of HpARI2 converts it from an inhibitor of signalling into a potentiator of IL-33 function[18]. Our structure rationalises why this is. By binding to IL-33 in a non-blocking manner, HpARI2_CCP1/2 stabilises the reduced, active form of the cytokine, extending its half-life in vivo, perhaps by limiting the conformational changes which would occur on oxidation and inactivation. Another *H. polygyrus bakeri*-derived inhibitor of IL-33 responses is HpBARI, which binds and blocks ST2, inhibiting IL-33 binding. Recently, oxidised IL-33 was shown to have ST2-independent effects, signalling via a RAGE-EGFR pathway, allowing it to act on ST2-negative epithelial cells[29]. No structural insights are currently available on the oxIL-33:RAGE:EGFR complex and no data has been obtained on whether HpARI2 binding to IL-33 could inhibit RAGE-EGFR signalling. As our previous findings indicate that HpARI2 can only bind to the reduced form of IL-33, while HpBARI binds only to ST2, it may be that oxIL-33:RAGE:EGFR responses on epithelial cells are unaffected during *H. polygyrus bakeri* infection, or may even be potentiated[15,16]. As epithelial cell responses are fundamentally altered during *H. polygyrus bakeri* infection, with suppression of tuft cell expansion and induction of a stem-like phenotype, this IL-33 pathway warrants further research[30].

Both inhibitors and potentiators of IL-33 have promising roles as therapeutics. For treatment of allergic diseases such as asthma, IL-33 blockade is advantageous. In contrast, IL-33 stabilisers which do not compete with ST2 could have uses in induction of parasitic worm expulsion, or reduction of metabolic dysfunction in obesity[31]. Our structural studies reveal the sites on IL-33 bound by both a potentiator (CCP2) and an inhibitor (CCP3) and will guide the future development of both types of IL-33-targeting therapeutic agents.

## Methods
### Ethics
The research complies with all relevant ethical regulations. Mouse experiments were performed under UK Home Office project licence

PP9520011, with institutional oversight performed by qualified veterinarians.

## Protein expression and purification

**HpARI2.** Mammalian expression constructs of HpARI2, its mutants and its truncated forms HpARI2_CCP1/2 and HpARI2_CCP2/3, each with a C-tag were cloned in the pHLsec vector. Expi293 F cells (Thermo Fisher) were transfected with expression constructs using the Expifectamine™ 293 Transfection Kit (Thermo Fisher) following the manufacturer's guidelines, including the addition of enhancers 1 and 2 (Thermo Fisher) 18 h post-transfection. Cell supernatants were harvested on the 4th-day post-transfection and passed through 0.4-micron filters. Supernatants were incubated with CaptureSelect™ C-tag affinity resin (Thermo Fisher) at 4 °C for 45 min. Following incubation, affinity resins were transferred to gravity flow columns and washed with 10-column volumes of 1 x HBS pH 7.2. Proteins were eluted using 2 M $MgCl_2$, 25 mM HEPES pH 7.2. Affinity-purified proteins were pooled and concentrated to 5 mg/ml and subjected to size exclusion chromatography (SEC) using an Superdex 75 10/300 increase column (cytiva) on an Äkta pure (GE Healthcare) into 150 mM NaCl, 50 mM HEPES, pH 7.2.

**Mouse IL-33.** IL-33 (residues 112–270) and its R169A mutant were cloned into vector pET28a (Invitrogen) and were expressed with a TEV protease cleavable N-terminal $His_6$ tag. The protein was expressed in *Escherichia coli* BL21 (DE3) induced with 0.5 mM IPTG at on $OD_{600}$ of 1.0–1.2 in overnight cultures grown at 17 °C. The cells were harvested by centrifugation at 4000 × g for 15 min and suspended with lysis buffer containing 20 mM Tris pH 8.0, 250 mM NaCl, 10% Glycerol and 10 mM imidazole, supplemented with a complete protease inhibitor cocktail (Roche). After sonication, the lysate was centrifuged at 50,000 × g for 45 min. The supernatant containing recombinant protein was then incubated with Ni-NTA resin for 45 min at 4 °C. After 20 column volume wash with lysis buffer, the protein was eluted using 20 mM HEPES pH7.2, 100 mM NaCl, 500 mM imidazole, 0.5 mM EDTA and 0.5 mM DTT. Purified protein was then mixed with TEV protease in a ratio of 50:1 and incubated overnight at room temperature to remove the his-tag, followed by SEC on an Superdex 75 10/300 column in 150 mM NaCl, 50 mM HEPES, pH 7.2.

**Mouse ST2.** The extracellular domain of mouse ST2 (residues Lys19 to Lys321) cloned into the pHLsec vector with a C-terminal $His_6$ tag and was expressed in Expi293 F cells (Thermo Fisher). Cell culture supernatants carrying expressed protein were harvested on the 4th-day post-transfection and filtered using the 0.45-micron filter. His-tagged ST2 was isolated by Ni-NTA chromatography, eluted with 500 mM imidazole, 150 mM NaCl, 50 mM HEPES, pH 7.2 and then further purified by SEC using the Superdex 200 10/300 increase column (cytiva) using 150 mM NaCl, 50 mM HEPES, pH 7.2.

## Surface plasmon resonance

**Assessing binding of HpARI2 and its variants to mIL-33.** HpARI2 variants, in 25 mM sodium phosphate pH 7.2, 150 mM NaCl, were biotinylated by mixing 50 μM of protein with 50 μM of EZ-Link™ Sulfo-NHS-Biotin (Thermo Fisher Scientific) followed by incubation on ice for 2 h. Excess biotin was removed using a PD-5 column equilibrated with 25 mM Tris pH 8.0, 150 mM NaCl. Experiments were performed at 25 °C on a Biacore T200 instrument (GE Healthcare) using the Biotin CAPture kit (cytiva) in 20 mM Tris-Cl pH 8.0, 150 mM NaCl, 1 mM EDTA, 0.02% Tween-20 and 1 mg/ml salmon sperm DNA. Purified analyte proteins were exchanged into this buffer using a PD-5 column. 300-400 RU of each HpARI2 variant was immobilised on the chip, and binding measurements were performed at a flow rate of 40 μl/min by injecting two-fold concentration series of IL-33 over the chip surface. For full-length HpARI2, a concentration range of 20 nM to 0.15 nM was

used, whereas 2 μM to 7.537 nM was used for HpARI2_CCP1/2 and 0.2 μM to 0.31 nM for HpARI2_CCP2/3. Association was measured for 400 s, followed by dissociation for 600 s. After each binding cycle, the sensor chip surface was regenerated by injecting 10 μl of 6 M guanidium-HCl and 1 M NaOH pH 11.0 mixed in a 4:1 ratio. The data was processed using BIA evaluation software version 1.0 (BIAcore, GE Healthcare). Response curves were double referenced by subtracting the signal from the reference cell and averaged blank injection. Binding kinetics and screening for various HpARI2 mutants and mIL-33R169A was done using same set of reagents and equipment.

**Assessing binding of FL-HpARI2 to hIL-33.** First, 400-450 RU of biotinylated HpARI2 was captured on the chip as above. A 2-fold dilution series (from 1000 nM to 15.62 nM) of hIL-33 or hIL-33:S158R was then injected over the chip surface to generate kinetic profiles for calculation of binding constants. After each binding cycle, the sensor chip surface was regenerated as above. Response curves were double referenced by subtracting the signal from the reference cell and averaged blank injection.

**Assessing binding of ST2-Fc to mIL-33.** To assess the direct impact of HpARI2 and its variants on mIL-33:ST2 interaction, 1000-1200 RU of ST2-Fc (R&D biosystems) was immobilised on a CM5 chip densely coated (5000-6000RU) with protein A/G using the standard amine coupling protocol. All proteins were first passed through the PD-5 column to buffer exchange into 20 mM Tris-Cl pH 8.0, 150 mM NaCl, 1 mg/ml salmon sperm DNA, 0.02% Tween-20. To measure competition, mIL-33 (2 μM) was mixed with 2.5 μM each of HpARI2, HpARI2_CCP2/3 and HpARI2-LD. The mixture was then flowed over an ST2-Fc coated chip, with an association time of 400 s and a dissociation time of 600 s. The sensor chip surface was regenerated by injecting 10 μl of 100 mM Glycine pH 1.5. Response curves were double referenced by subtracting the signal from the reference cell and averaged blank injection.

## Crystallisation of the HpARI2_CCP2/3:mIL-33 complex

Putative N-glycosylation sites on HpARI2 (residues N113 and N128) were removed by N to Q mutations, and non-glycosylated HpARI2_CCP2/3 was expressed and purified as above. Purified HpARI2_CCP2/3 was mixed with a 1.5-fold molar excess of mIL-33 and incubated for 15 min on ice concentrated to 500 μl and injected into an S75 10/300 increase (cytiva) column equilibrated with 10 mM HEPES pH 7.2, 100 mM NaCl, 0.1 mM DTT. The peak corresponding to the complex was concentrated to 15 mg/ml for crystallisation screens. Crystals were obtained within 48 h, with an optimised condition of 0.1 M Bis-Tris pH 6.8 and 22% PEG3350. Crystals were cryo-protected using 0.1 M Bis-Tris pH 6.8, 22% PEG3350, 20% ethylene glycol and cryo-cooled by direct plunging into liquid nitrogen.

## Crystallographic structure determination

Diffraction data were collected at Diamond light Source beamline IO4. All data were integrated and scaled using the DIALS[32] with 8% of reflections set aside for $R_{free}$. The structures were determined by maximum-likelihood molecular replacement (MR) implemented in the programme suite PHASER[33]. Search models for mIL-33 were derived from PDB 5VI4 (IL33:ST2:IL-1RAcP)[9], and the search template for HpARI2_CCP2/3 was obtained using Alphafold2[9,34]. In both cases, side chains were pruned to alanine and flexible regions such as loops were deleted from search models before molecular replacement. Model building was performed in COOT[35], and individual coordinate and ADP refinement combined with TLS was performed in phenix.refine[36] and Buster[37]. Model and map validation tools in COOT, the PHENIX suite and the PDB validation server were used throughout the workflow to guide improvement and validate the quality of crystallographic models[38].

## SAXS analysis of HpARI2 variants and their complex with IL-33

SEC SAXS experiments were carried out at the B21 beamline at Diamond Light Source, using X-rays at a wavelength of 0.99 Å and an Eiger 4 M detector (Dectris). For data collection, samples were concentrated and injected at 20 °C to a Superdex 200 Increase 3.2/300 column equilibrated with 20 mM Tris-Cl, 300 mM NaCl, pH 7.2, with a 2 s exposure for each frame. The data were processed in bioXTAS-RAW[39] or chromixs[40] by first averaging and subtracting buffer frames from averaged frames corresponding to peak fractions[41] and were then processed using the ATSAS package[42]. The distance distribution function P(r) and the maximum particle diameter $D_{max}$ were determined using GNOM[43]. For volumetric reconstructions ab initio bead models were generated using DAMMIF[44]. These models were then averaged with DAMAVER[42,45], then refined against the experimental data using DAMMIN[46]. The resulting bead models were then used to draw volumetric envelopes with Situs[47]. Crystal structures of HpARI2_CCP2/3:mIL-33 complexes and Alphafold2 models for FL-HpARI2 and FL-HpARI2:mIL-33 were docked into the envelopes using Chimera[48] and fitted to experimental scattering data using Crysol[45,49].

## Circular Dichroism

Circular dichroism was performed with a Jasco J- 815 spectropolarimeter (Jasco, Japan). Spectra were obtained in duplicate with 0.5 nm step resolution, a response time of 1 s and a scanning speed of 10 nm/min. The spectrum for each HpARI2 and its glycosylated versions was recorded from 190 to 250 nM at a protein concentration of 0.01 mg/ml in 0.1 mm quartz cuvette. The baseline (20 mM phosphate buffer pH 7.2 and 25 mM NaF) was subtracted from all measurements.

## In vitro bone-marrow derived ILC2 assay

Bone marrow single cell suspensions were prepared from three 6–10 week old male C57BL/6 J mice (Charles River), by flushing tibias and femurs with RPMI 1640 medium using a 21 g needle. Cells were resuspended in ACK lysis buffer (Gibco) for 5 min at room temperature, prior to resuspension in complete RPMI (with 10% FCS, 1% Penicillin/Streptomycin, 1% L-glutamine, Gibco) and passing through a 70 μm cell strainer. Cells were cultured in round-bottom 96-well-plates in a final 200 μl volume, containing $0.5 \times 10^6$ cells/well. IL-2 and IL-7 (Biolegend) were added at 10 ng/ml final concentration, IL-33 (Biolegend) at 1 ng/ml final concentration. HpARI2 truncations or N-glycosylated variants were added in a range of concentrations. Cells were then cultured at 37 °C, 5% $CO_2$ for 5 days, prior to collection of supernatants. IL-5 and IL-13 concentration were assessed following manufacturer's instructions using mouse uncoated IL-5 and IL-13 ELISA kits (Invitrogen).

## In vivo Alternaria model

Female BALB/cAnNCrl mice were purchased from Charles River, UK. Mice were accommodated and procedures performed under UK Home Office licences with institutional oversight performed by qualified veterinarians. Home office project licence PP9520011. At 6–10 weeks of age, mice were intranasally administered with 50 μg Alternaria allergen (Greer XPM1D3A25) and 10 μg of HpARI2 or N-glycosylated HpARI2 variants suspended in PBS, all carried out under isoflurane anaesthesia. Two repeat experiments were carried out, the first with four mice per group (Alt, Alt + HpARI2 and Alt + HpARI2_N70 and Alt + HpARI2_LD groups only), the second with three mice per group (PBS, Alt, Alt + HpARI2 and Alt + HpARI2_N70 and Alt + HpARI2_LD groups). Data was pooled to give a total of n = 7 for all groups except PBS control, which was n = 3. Mice were culled 24 h later, and lungs were taken for single-cell preparation and flow cytometry[16]. Lungs were digested in PBS containing 2 U/ml of Liberase TL (Roche) and 80 U/ml DNase (Thermo Fisher) shaking at 37 °C for 35 min. Digested tissue was further macerated through a 70 μm cell strainer (Greiner Bio-One), and red blood cells lysed using ACK buffer (Thermo Fisher).

A haemocytometer with trypan blue was used to count live cells. Single cell suspensions were washed in PBS and stained with Fixable Blue Live/Dead stain (Thermo Fisher) according to manufacturer instructions. Cells were then blocked with anti-mouse CD16/32 antibody (Biolegend, clone 93, #101302) and stained with CD45-AlexaFluor700 (Biolegend, clone 30-F11, #103128), Siglecf-PE (Miltenyi, clone ES22-10D8, #130-102-274) and CD11c-AlexaFluor647 (Biolegend, clone N418, #117312). Samples were acquired on an LSR Fortessa (BD Biosciences) and analysed using FlowJo 10 (Treestar). Lung eosinophils were identified as SiglecF$^{hi}$CD11c$^-$CD45$^+$ live cells.

## Statistics and reproducibility

Sample sizes are described in figure legends and methods. No statistical method was used to predetermine sample size. Quantitative experiments were typically repeated in technical triplicate. No data were excluded from the analyses. The experiments were not randomized. The Investigators were not blinded to allocation during experiments and outcome assessment.

## Reporting summary

Further information on research design is available in the Nature Portfolio Reporting Summary linked to this article.

## Data availability

Data within graphs and uncropped gel and blot images are included as a source data file. Crystallographic data generated in this study have been deposited in the protein data bank with accession code 8Q5R (HpARI:IL33). Crystallographic data used in this study are from the protein data bank, with accession codes 5VI4 (IL33:ST2:IL-1RAcP) and 2KLL (IL33). SAXS data generated in this study have been deposited in SASBDB with accession codes SASDUW6, SASDUX6, SASDUY6 and SASDUZ6. Source Data is provided with this paper. Source data are provided with this paper.

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

## Acknowledgements

This work was funded by a Wellcome Investigator Award (221914/Z/20/Z) and was supported by LONGFONDS Accelerate as part of the AWWA project. The authors would like to thank Dr Ed Lowe and the beamlines scientists at Diamond beamline I03 for help with crystallographic data collection and Danielle Smyth for help with logistics and protein expression. We thank Dr. Nikul Khunti (B21, Diamond light source) for help with SEC-SAXS data collection.

## Author contributions

A.J., H.J.M. and M.K.H. conceived and planned the study and wrote the manuscript. A.J. conducted protein production, interaction analysis, biophysics studies ad crystallographic structure determination. F.C. conducted in vitro signalling assays. F.C. and H.J.M. conducted mouse experiments. All authors designed experiments, analysed data, and read and commented on the manuscript.

## Competing interests

The authors declare no competing interests.
