## [Peer Review File · Nature Communications]

REVIEWER COMMENTS

Reviewer #1 (Remarks to the Author):

The manuscript by Abhishek Jamwal et al. discusses their findings on the crystal structure of the HpARI_CCP2/3 complexed with mIL-33, which revealed that HpARI inhibits IL-33-mediated signaling by directly blocking ST2 binding. The authors first used domain truncation, surface plasmon resonance (SPR), and cellular assays to demonstrate that the CCP1 domain of HpARI does not contribute to its effector function. Next, the researchers determined the crystal structure of the HpARI_CCP2/3 : mIL-33 complex at a high resolution, allowing them to uncover the atomic details of the IL-33-binding interfaces on CCP2 and CCP3. They also found that a long loop in the CCP3 domain plays a critical role in mediating the inhibition of IL-33 signaling by HpARI. This was confirmed through additional SPR and cell-based assays in vitro, as well as a mice model in vivo. Overall, these findings shed light on the mechanism by which HpARI inhibits IL-33 signaling and provide the first structure of IL-33 bound to a pathogen-based inhibitor. This knowledge may be valuable in guiding the development of future therapeutic agents targeting IL-33.

The questions and suggestions include:

(1) The authors introduced N-glycosylation sites at positions N69, N70, and N152 to determine the binding interface of HpARI_CCP2/3:mIL-33 in different molecular packings in the crystal structure. The introduction of N-glycans is uncommon, and the mutations were justified by SDS-PAGE gel, gel-filtration profile, and CD measurement. Please provide the sequences of the "NXS/T" motif introduced at positions 69, 70, and 152.

(2) "We then assessed the importance of this interaction by deleting residues 180-188 from full-length HpARI (HpARI-LD), which would sufficiently truncate the loop to prevent it from reaching its binding site." The LD mutant was constructed by deleting a long loop from 180 to 188. This LD mutant is important because it was used in subsequent in vitro and in vivo studies. The authors should present the SDS-PAGE gel, gel-filtration profile, and CD measurement of the LD mutants, as they did for the N-glycosylation mutants.

(3) This question is still regarding the LD mutant. At the tip end of the 180-188 loop, there is a short helix that contacts mIL-33, and "Interactions with IL-33 are mostly formed by non-polar residues from, and C-terminal to, a small helix which docks into a small concave, hydrophobic pocket on the top of the mIL-33 b-barrel." Would residues 180, 184, 185, 186, 187, and 188 of HpARI be considered hot-spot residues for inhibition? Did the authors explore single-site mutations or a combination of single-site mutations instead of directly deleting the 180-188 loop?

(4) This question pertains to the deleted CCP1 domain in the crystal structure. I agree with the authors that CCP2 and CCP3 are the most important regions for the function of HpARI. However, truncation of CCP1 resulted in a significant reduction in affinity and increased concentrations required for inhibition of IL-33 function by nearly a hundred-fold (KD ~48pM to ~4 nM, IEC50 40 pM to 3 nM). This indicates that

CCP1 still plays a role in binding. In the SAXS analysis, the authors still used HpARI_CCP2/3. I wonder if using HpARI including CCP1 in the SAXS study would provide hints about the role of CCP1 in the binding.

(5) The resolution cutoff was set at 1.93 angstroms. However, it appears that the authors were overly ambitious in pursuing high resolution. Important statistics for resolution cutoff in the outmost shell in the extended data table 2, including Rpim, CC1/2, and I/sigma, do not support a resolution cutoff of 1.93 angstrom.

(6) Extended Data Figure 3 and Extended Data Table 1: mL-33R R169A should be R159A.

Reviewer #2 (Remarks to the Author):

Authors present the structure of HpARI bound to mouse IL-33 at high 1.9Å resolution. The mutational study defined the most critical residues between the HpARI and IL33. Using solution X-ray scattering, authors identify the biological relevant unit that was further confirmed SPR and relevant mutation in the HpARI-CCP2 domain or deletion of HpARI-CCP3 log loop, respectively. Authors also show that truncations of HpARI, which lack the large loop from CCP3

are not able to block IL-33-mediated signaling in a cell-based assay and in an in vivo model of

asthma. Indeed, the CCP3 long loop that forms additional interaction with IL33 is one of the kinds of interaction that should be considered in future therapeutic design. The manuscript is well written, and experimental structural studies, in-vitro and in-vivo assay support the conclusion that direct competition between HpARI and ST2 is responsible for suppressing IL-33-dependent responses. However, to be able to comment on structural allostery in ST2 - IL33 and ST-HpARI complex, I wonder about the "ground state" of HpARI in the absence of IL33.

It is well established that ST2 binds IL33 in an allosteric manner. The ST2 D2-D3 was wrapping around IL33. This allosteric transition of ST2 allows further interaction with its co-receptor. Please provide evidence that the discussed lock-in key interaction between HpARI_CCP2-CCP3 and IL33 works in the same or different mechanism. It will also be interesting to know whether CCP3 long loop is solvent-exposed and flexible in solution before its interaction with IL33 or well-folded, as shown in its complex structure. I ask to extend the solution scattering study and provide insight into HpARI CCP2-CCP3 flexibility using SAXSD-based multi-state modeling.

More technical concerns:

Provide residuals to the experimental/theoretical profile fit.

Deposit the SAXS data in the relevant SAXS database.

Provide the Guinier plot in the extended Figure 2 for further SAXS data validations.

Provide SEC-SAXS chromatogram in extended figure.

Use the full sequence of (filled up the missing loops and glycans) in your atomistic SAXS fitting.

Explain here presented the so-called "chicken-drum" stick ab initio SAXS model. It's due to aggregation or flexible CCP3-loop, or the presence of glycans. Avoid SAXS shape modeling in general in case you provide an atomistic model.

Reviewer #3 (Remarks to the Author):

The manuscript by Jamwal et al. decisively builds upon the authors' pioneering discovery of HpARI as a helminth protein that suppresses the activity of the pro-inflammatory cytokine IL-33 in the mouse. The current study reports fascinating structural insights into the very unusual structure of HpARI and how it might sequester the activity of mouse IL-33 by competing for binding to mouse IL-33 against the cognate high-affinity receptor ST2. Although in a first instance this work will contribute to a better understanding of the activity of IL-33 in mouse models of allergy and inflammation and the general principles of host immune system evasion by parasites, this work has clear implications for further interrogating IL-33 signaling in a more general context including the human counterpart.

I herewith provide the following input for the authors' consideration:

Structure-based sequence alignments of mouse vs human IL-33 will be needed to document possible insights as to protein design possibilities for human IL-33.

In this regard, the authors allude to such perspectives arising from their study but actually do not discuss this at all in any concrete way.

What is the cross-reactivity of human IL-33 with HpARI? Would it be possible to provide some data on this, perhaps supported by appropriately annotated structure-based sequence alignments (see previous point).

The approach to introduce N-linked glycosylation sites at the three putative interaction interfaces via mutagenesis to probe the relevance of the related interfaces is elegant. However, while the data shown in the SDS-PAGE analyses in Extended Figure 2 do suggest that those sites were indeed glycosylated, the better way to demonstrate this would be to analyze by SDS-PAGE protein samples treated with PNGaseF.

This would additionally serve to set the baseline for the electrophoretic mobility of the WT protein in the first place.

It would be informative to provide up front, BSA values for all three candidate interaction interfaces derived from the crystal structure and lattice contacts.

The SAXS data presented in Ext. Data Fig. 2e could be considered as rather inconclusive given the small differences in the calculated χ^2 values and the fact that glycans were not modelled. This reviewer's experience strongly suggests that modelling a single glycan can drastically shift such χ^2 values.

Furthermore, the SAXS data is not reported appropriately via a dedicated table, analogous to the crystallographic data/refinement reporting.

To this end, the authors are strongly advised to consult Trewhella et al. 2023 DOI: 10.1107/S2059798322012141 for how to report the SAXS data.

The reporting of binding studies via SPR is only expressed in terms of affinities in the main text, while the binding kinetics reported via the Extended Data Table do have more information content in terms of the on- and off-rates of the studied interactions. Some of the affinities reported are characterized by very fast on-rates (as fast as 10^6 /Ms) or very slow off-rates (as slow as 10^{-5} /s).

Make sure to adopt accurately the nomenclature for binding constants, with *K* or *k* in italics and all subscripts as normal.

The competitive binding of ST2 and HpARI to mIL-33 is not addressed rigorously. Given the importance of this competitive binding model for the mode of action of HpARI this interaction needs to be characterized more accurately and thoroughly. For instance, why was IL-1RAcP not included in these experimental undertakings.

This reviewer would like to suggest expanding these undertakings to include IL-1RAcP and to move the new data to a main display item.

In addition, the structural models for the competitive binding mode would be more informative as part of a main display item, most preferably with data from the herein proposed experimental undertakings.

In this regard, annotated structure-based sequence alignments facilitated by perhaps predicted models for the assembly of the mIL-33:ST2:IL-1RAcP model could provide delineation of the possible degree of overlap between the respective competitive binding sites.

Extended Data Figure 3 would be more appropriate for a main display item.

Extended Data Table 2:

Report number of reflections used for R-free.

Specify in the table legend what the numbers in parentheses correspond to.

Extended Data Figure 2f: Report rmsd values.

Extended Data Figure 1: A label for the molecular weight markers in kDa is missing.

REVIEWER COMMENTS

Reviewer #1 (Remarks to the Author):

The manuscript by Abhishek Jamwal et al. discusses their findings on the crystal structure of the HpARI_CCP2/3 complexed with mIL-33, which revealed that HpARI inhibits IL-33-mediated signaling by directly blocking ST2 binding. The authors first used domain truncation, surface plasmon resonance (SPR), and cellular assays to demonstrate that the CCP1 domain of HpARI does not contribute to its effector function. Next, the researchers determined the crystal structure of the HpARI_CCP2/3 : mIL-33 complex at a high resolution, allowing them to uncover the atomic details of the IL-33-binding interfaces on CCP2 and CCP3. They also found that a long loop in the CCP3 domain plays a critical role in mediating the inhibition of IL-33 signaling by HpARI. This was confirmed through additional SPR and cell-based assays *in vitro*, as well as a mice model *in vivo*. Overall, these findings shed light on the mechanism by which HpARI inhibits IL-33 signaling and provide the first structure of IL-33 bound to a pathogen-based inhibitor. This knowledge may be valuable in guiding the development of future therapeutic agents targeting IL-33.

We thank the reviewer for their balanced and positive assessment of the work.

The questions and suggestions include:

(1) The authors introduced N-glycosylation sites at positions N69, N70, and N152 to determine the binding interface of HpARI_CCP2/3:mIL-33 in different molecular packings in the crystal structure. The introduction of N-glycans is uncommon, and the mutations were justified by SDS-PAGE gel, gel-filtration profile, and CD measurement. Please provide the sequences of the "NXS/T" motif introduced at positions 69, 70, and 152.

We have now added this into Extended Data Figure 2b.

(2) "We then assessed the importance of this interaction by deleting residues 180-188 from full-length HpARI (HpARI-LD), which would sufficiently truncate the loop to prevent it from reaching its binding site." The LD mutant was constructed by deleting a long loop from 180 to 188. This LD mutant is important because it was used in subsequent *in vitro* and *in vivo* studies. The authors should present the SDS-PAGE gel, gel-filtration profile, and CD measurement of the LD mutants, as they did for the N-glycosylation mutants.

We have now added this data into Extended Data Figure 2. The SDS PAGE is in panel c) and the gel filtration profile and CD measurements are in panel d)

(3) This question is still regarding the LD mutant. At the tip end of the 180-188 loop, there is a short helix that contacts mIL-33, and "Interactions with IL-33 are mostly formed by non-polar residues from, and C-terminal to, a small helix which docks into a small concave, hydrophobic pocket on the top of the mIL-33 b-barrel." Would residues 180, 184, 185, 186, 187, and 188 of HpARI be considered hot-spot residues for inhibition? Did the authors explore single-site mutations or a combination of single-site mutations instead of directly deleting the 180-188 loop?

The interaction between the HpARI2 loop and IL-33 is predominantly mediated by four hydrophobic residues from HpARI2. These come together to form a hydrophobic patch which docks into a hydrophobic pocket on IL-33. We therefore think that it is unlikely that one or two residues act as hot-spots or that single point mutations will substantially disrupt binding in this region. We therefore prefer not to conduct a mutagenesis screen of this region of the protein. Indeed, this contact surface is extremely well-resolved and unambiguous in the electron density and it is our view that no further validation is required of this part of the interaction.

(4) This question pertains to the deleted CCP1 domain in the crystal structure. I agree with the authors that CCP2 and CCP3 are the most important regions for the function of HpARI. However, truncation of CCP1 resulted in a significant reduction in affinity and increased concentrations required for inhibition of IL-33 function by nearly a hundred-fold (KD ~48pM to ~4 nM, IEC50 40 pM to 3 nM). This indicates that CCP1 still plays a role in binding. In the SAXS analysis, the authors still used HpARI_CCP2/3. I wonder if using HpARI including CCP1 in the SAXS study would provide hints about the role of CCP1 in the binding.

We have now also collected SAXS data for the HpARI2, both alone and also bound to IL-33 and this is presented in Extended Data Figure 3 and is also described in lines 118-124. This shows that the three domains of HpARI2 form a linear and relatively rigid arrangement. We see no clear direct interaction between CCP1 and IL-33. Instead, CCP1 appears to make a substantial interface with CCP2 and our hypothesis is that the presence of CCP1 stabilises CCP2 in a conformation which binds to IL-33. The data is not conclusive, but we have made this speculation in lines 145-150.

(5) The resolution cutoff was set at 1.93 angstroms. However, it appears that the authors were overly ambitious in pursuing high resolution. Important statistics for resolution cutoff in the outmost shell in the extended data table 2, including Rpim, CC1/2, and l/sigma, do not support a resolution cutoff of 1.93 angstrom.

We agree with the review that we pushed this data a little too far. We have now processed instead with a cut-off of 2.1Å resolution. This results in more appropriate statistics, including a CC1/2 in the outer shell of 0.723.

(6) Extended Data Figure 3 and Extended Data Table 1: mL-33R R169A should be R159A.

These corrections have been made (now Extended Data Figure 4).

Reviewer #2 (Remarks to the Author):

Authors present the structure of HpARI bound to mouse IL-33 at high 1.9Å resolution. The mutational study defined the most critical residues between the HpARI and IL33. Using solution X-ray scattering, authors identify the biological relevant unit that was further confirmed SPR and relevant mutation in the HpARI-CCP2 domain or deletion of HpARI-CCP3 log loop, respectively. Authors also show that truncations of HpARI, which lack the large loop from CCP3 are not able to block IL-33-mediated signaling in a cell-based assay and in an in vivo model of asthma. Indeed, the CCP3 long loop that forms additional interaction with IL33 is one of the kinds of interaction that should be considered in future therapeutic design. The manuscript is well written, and experimental structural studies, in-vitro and in-vivo assay support the conclusion that direct competition between HpARI and ST2 is responsible for suppressing IL-33-dependent responses.

We thank for the reviewer for this positive response to our work.

However, to be able to comment on structural allostery in ST2 - IL33 and ST-HpARI complex, I wonder about the "ground state" of HpARI in the absence of IL33. It is well established that ST2 binds IL33 in an allosteric manner. The ST2 D2-D3 was wrapping around IL33. This allosteric transition of ST2 allows further interaction with its co-receptor. Please provide evidence that the discussed lock-in key interaction between HpARI_CCP2-CCP3 and IL33 works in the same or different mechanism. It will also be interesting to know whether CCP3 long loop is solvent-exposed and flexible in solution before its interaction with IL33 or well-folded, as shown in its complex structure. I ask to extend the solution scattering study and provide insight into HpARI CCP2-CCP3 flexibility using SAXSD-based multi-state modeling.

Firstly, we note that the aim of our manuscript is not to reveal the structural allostery of these protein complexes but to show how HpARI2 binds to IL-33 and inhibits IL-33 from interacting with ST2. Had we wished to investigate the structural dynamics of HpARI2, we would have selected a different set of methods, such as NMR and molecular dynamics simulations. We have not made claims about the dynamics of the interaction, which lies outside the scope of the manuscript.

Nevertheless, we have now included additional SAXS data (Extended Data Figure 3). We collected x-ray scattering data for HpARI2-CCP2/3 and FL_HpARI2, both alone and in complex with IL-33. Our conclusion is that the three domains of HpARI2 form an elongated arrangement without substantial flexibility, which docks directly onto IL-33. Therefore, unlike ST2, which wraps around IL-33, HpARI2 appears to be mostly unchanged in conformation on binding. This is described in lines 118-124.

We have not investigated the degree of flexibility of the long loop. In the IL-33-bound conformation, this adopts a well-defined structure in the electron density. Understanding its degree of flexibility when not bound to IL-33 is outside the scope of this study and not a question which we plan to explore.

More technical concerns:

Provide residuals to the experimental/theoretical profile fit.

Deposit the SAXS data in the relevant SAXS database.

Provide the Guinier plot in the extended Figure 2 for further SAXS data validations.

Provide SEC-SAXS chromatogram in extended figure.

Use the full sequence of (filled up the missing loops and glycans) in your atomistic SAXS fitting.

Explain here presented the so-called "chicken-drum" stick ab initio SAXS model. It's due to aggregation or flexible CCP3-loop, or the presence of glycans. Avoid SAXS shape modeling in general in case you provide an atomistic model.

While SAXS data formed a very minor part of the initial submission of this paper, we now present more SAXS data and have increased the degree to which we describe the data in the paper. We have provided all of the information requested in Extended Data Figure 3 and Extended Data Table 3.

Reviewer #3 (Remarks to the Author):

The manuscript by Jamwal et al. decisively builds upon the authors' pioneering discovery of HpARI as a helminth protein that suppresses the activity of the pro-inflammatory cytokine IL-33 in the mouse. The current study reports fascinating structural insights into the very unusual structure of HpARI and how it might sequester the activity of mouse IL-33 by competing for binding to mouse IL-33 against the cognate high-affinity receptor ST2. Although in a first instance this work will contribute to a better understanding of the activity of IL-33 in mouse models of allergy and inflammation and the general principles of host immune system evasion by parasites, this work has clear implications for further interrogating IL-33 signaling in a more general context including the human counterpart.

We thank the reviewer for their positive assessment of the work and its implications.

I herewith provide the following input for the authors' consideration:

Structure-based sequence alignments of mouse vs human IL-33 will be needed to document possible insights as to protein design possibilities for human IL-33.

In this regard, the authors allude to such perspectives arising from their study but actually do not discuss this at all in any concrete way.

We now include these sequence alignments in Extended Data Figure 5

What is the cross-reactivity of human IL-33 with HpARI? Would it be possible to provide some data on this, perhaps supported by appropriately annotated structure-based sequence alignments (see previous point).

We have conducted an analysis of the binding of human IL-33 to HpARI2 and present this, together with a sequence alignment in Extended Data Figure 5 and a description in lines 151-160 and 173-177. HpARI2 does bind to human IL-33 ($K_D=65\text{nM}$) and a single point mutation (S158R), guided by the sequence alignment and crystal structure of the house homologue, increases the binding affinity nearly 100-fold. In addition, the effect of glycosylation and loop deletion mutations is the same on human and mouse IL-33 binding. Therefore, while this is not relevant to the natural function of HpARI2 it does show that human and mouse IL-33s bind to HpARI2 with the same binding more and does support possible use of HpARI2 variants to regulate human IL-33 function.

The approach to introduce N-linked glycosylation sites at the three putative interaction interfaces via mutagenesis to probe the relevance of the related interfaces is elegant. However, while the data shown in the SDS-PAGE analyses in Extended Figure 2 do suggest that those sites were indeed glycosylated, the better way to demonstrate this would be to analyze by SDS-PAGE protein samples treated with PNGaseF. This would additionally serve to set the baseline for the electrophoretic mobility of the WT protein in the first place. It would be informative to provide up front three candidate interaction interfaces derived from the crystal structure and lattice contacts.

We have conducted the experiment proposed by the reviewer and have provide a gel of the HpARI2 mutants after denaturation and deglycosylation with PNGaseF. This is shown in Extended Data Figure 2c. This confirms that the N69, N70 and N152 mutations do not change the mobility of HpARI2, and that the mobility change in the non-PNGaseF treated samples are due to glycosylation. The two interfaces tested are shown in Extended Data Figure 2a.

The SAXS data presented in Ext. Data Fig. 2e could be considered as rather inconclusive given the small differences in the calculated χ^2 values and the fact that glycans were not modelled. This reviewer's experience strongly suggests that modelling a single glycan can drastically shift such χ^2 values.

Furthermore, the SAXS data is not reported appropriately via a dedicated table, analogous to the crystallographic data/refinement reporting.

To this end, the authors are strongly advised to consult Trehwella et al. 2023 DOI: 10.1107/S2059798322012141 for how to report the SAXS data.

On reflection, we agree with the reviewer that the small change in χ^2 is not sufficient to convincingly determine which of the crystallographic contacts represent the biological complex. We have therefore removed this claim from the manuscript and instead rely on mutagenesis data (both glycan insertion and the effect of mutation of the long loop) to confirm the biologically active complex.

We do now include a broader panel of SAXS data Extended Data Figure 3 to address questions of protein dynamics and have reported this SAXS data more comprehensively.

The reporting of binding studies via SPR is only expressed in terms affinities in the main text, while the binding kinetics reported via the Extended Data Table do have more information content in terms of the on- and off-rates of the studied interactions. Some of the affinities reported are characterized by very fast on-rates (as fast as 10^6 /Ms) or very slow off-rates (as slow as 10^{-5} /s). Make sure to adopt accurately the nomenclature for binding constants, with K or k in italics and all subscripts as normal.

We thank the reviewer for noticing that all of the SPR data is fully reported in the tables. We have checked that our reporting of constants is consistent and expect that these will be put into the journal style during editing. We have also described off rates where we think that they are biologically important (i.e. line 82).

The competitive binding of ST2 and HpARI to mIL-33 is not addressed rigorously. Given the importance of this competitive binding model for the mode of action of HpARI this interaction needs to be characterized more accurately and thoroughly. For instance, why was IL-1RAcP not included in these experimental undertakings. This reviewer would like to suggest expanding these undertakings to include IL-1RAcP and to move the new data to a main display item.

In addition, the structural models for the competitive binding mode would be more informative as part of a main display item, most preferably with data from the herein proposed experimental undertakings.

In this regard, annotated structure-based sequence alignments facilitated by perhaps predicted models for the assembly of the mL-33:ST2:IL-1RAcP model could provide delineation of the possible degree of overlap between the respective competitive binding sites.

We do not agree with the reviewer about these studies. Our paper is about the competition between HpARI2 and ST2 for the binding to IL-33 and we show data to confirm this completion in Figure 3. This includes a structural overlay of the IL-33:ST2 and IL-33:HpARI2 complex structures to show how this competition works.

IL-1RAcP binds to ST2 only when ST2 is bound to IL-33 and it has been shown in published studies that free ST2 does not bind to IL-1RAcP. We show that HpARI2 binds to IL-33 and this prevents IL-33 from binding to ST2 through a direct competition. It is therefore self-evident that this leads to free ST2 which cannot bind to IL-1RAcP. There is no binding of HpARI2 to either ST2 or to IL-1RAcP and so no mechanism through which HpARI2 can modulate IL-1RAcP signalling other than by blocking IL-33 from binding to ST2. It is therefore not clear to us what experiments we could do that would add to this model involved IL-1RAcP.

Extended Data Figure 3 would be more appropriate for a main display item.

In our view data provides a validation of the structure and the interaction, rather than a biologically important finding and we therefore prefer to retain it as an Extended Data item.

Extended Data Table 2:

Report number of reflections used for R-free.

We have used the convent of setting aside 8% of reflections for Rfree, and now mention this in line 352.

Specify in the table legend what the numbers in parentheses correspond to.

We have made this addition.

Extended Data Figure 2f: Report rmsd values.

We have added this, now Extended Data Figure 3e.

Extended Data Figure 1: A label for the molecular weight markers in kDa is missing.

We have added this.

REVIEWERS' COMMENTS

Reviewer #1 (Remarks to the Author):

The authors have provided additional data and revised the manuscript to answer my questions and suggestions. I suggest that the revised manuscript is accepted for publication.

Reviewer #2 (Remarks to the Author):

The revision of Jamwal et al. has addressed all my concerns about the original manuscript. The authors have done a diligent job in their revisions to answer queries and correct any mistakes. The additional SAXS experimentation was helpful. Thank you for depositing SAXS data to SASBDB.

Reviewer #3 (Remarks to the Author):

The authors' revised manuscript is a substantial improvement over the originally submitted manuscript.

Nevertheless, I think that the way the following point was addressed is still inadequate:

" The competitive binding of ST2 and HpARI to mL-33 is not addressed rigorously. Given the importance of this

competitive binding model for the mode of action of HpARI this interaction needs to be characterized more

accurately and thoroughly. For instance, why was IL-1RAcP not included in these experimental undertakings."

I do agree that this point does not need to be addressed experimentally given the mechanistic requirements of the receptor assembly mediated by IL-33 and the clear competitive binding of HpARI2 to IL-33 compared to ST2. However, the mechanistic considerations the authors provided in their rebuttal need to be included in the manuscript via a short discussion and relevant referencing of key literature, perhaps most appropriately, in the section at line 183 "HpARI2 blocks IL-33 from binding it receptor ST2...".